# Exosomes as an Emerging Plasmid Delivery Vehicle for Gene Therapy

**DOI:** 10.3390/pharmaceutics15071832

**Published:** 2023-06-27

**Authors:** Margaret Wallen, Farrukh Aqil, Wendy Spencer, Ramesh C. Gupta

**Affiliations:** 13P Biotechnologies, Inc., Louisville, KY 40202, USA; margaret.3pbiotech@gmail.com (M.W.); wendy.spencer3p@gmail.com (W.S.); 2Brown Cancer Center, University of Louisville, Louisville, KY 40202, USA; farrukh.aqil@louisville.edu; 3Department of Medicine, University of Louisville, Louisville, KY 40202, USA; 4Department of Pharmacology and Toxicology, University of Louisville, Louisville, KY 40202, USA

**Keywords:** exosomes, nanoplatform, plasmid DNA delivery, cancer, gene therapy

## Abstract

Despite its introduction more than three decades ago, gene therapy has fallen short of its expected potential for the treatment of a broad spectrum of diseases and continues to lack widespread clinical use. The fundamental limitation in clinical translatability of this therapeutic modality has always been an effective delivery system that circumvents degradation of the therapeutic nucleic acids, ensuring they reach the intended disease target. Plasmid DNA (pDNA) for the purpose of introducing exogenous genes presents an additional challenge due to its size and potential immunogenicity. Current pDNA methods include naked pDNA accompanied by electroporation or ultrasound, liposomes, other nanoparticles, and cell-penetrating peptides, to name a few. While the topic of numerous reviews, each of these methods has its own unique set of limitations, side effects, and efficacy concerns. In this review, we highlight emerging uses of exosomes for the delivery of pDNA for gene therapy. We specifically focus on bovine milk and colostrum-derived exosomes as a nano-delivery “platform”. Milk/colostrum represents an abundant, scalable, and cost-effective natural source of exosomes that can be loaded with nucleic acids for targeted delivery to a variety of tissue types in the body. These nanoparticles can be functionalized and loaded with pDNA for the exogenous expression of genes to target a wide variety of disease phenotypes, overcoming many of the limitations of current gene therapy delivery techniques.

## 1. Introduction

According to the US Food and Drug Administration (FDA), the basic concept of gene therapy is to introduce exogenous genetic material to alter gene expression for the purpose of ameliorating disease symptoms [1]. Depending on the nature of the expression abnormality, this can be accomplished in multiple ways; first, a defective or missing gene may be replaced; second, an overly active gene may be suppressed; or third, an entirely novel gene may be introduced [1,2]. The theoretical framework for gene therapy began in the late 1960s and early 1970s [3], even before the human genome had been sequenced, and the age of “omics” has only expanded science’s understanding of diseases with genetic underpinnings, allowing for the identification of additional therapeutic targets.

The idea for gene therapy originated based on observations of natural occurrences of modifications of gene expression [4]. The classic Avery, MacLeod, and McCarty experiment in 1944 concluded that foreign DNA, the transforming principle, could be engulfed by bacterial cells and alter their physical traits and pathogenicity [5]. The discovery of a natural transformation system by viruses creating stable alterations in the genotypes of the cells they infected evoked the idea of engineering the same infectious agents to purposefully introduce exogenous genes for therapeutic benefit [4]. 

Early attempts were unsuccessful, including an attempt to deliver the unmodified Shope virus, which naturally induces arginase expression, to treat hyperargininemia [6]. The public at large was not receptive to the idea of purposefully introducing viruses into human cells or the human body, and progress to approved clinical trials was slow [7]. In 1989, the National Institutes of Health (NIH) and the FDA approved the first introduction of foreign genes into humans, although not for therapeutic purposes, to provide proof-of-concept for the expression of exogenous genetic material [8]. The first approved therapeutic procedure came the next year, the successful infusion of genetically modified autologous T cells to treat Severe Combined Immunodeficiency (SCID) [9].

The next 10 years, from 1989 to 1999, were filled with optimism and many investigators applied for phase I and phase II clinical trials [10]. In 1999, the death of 18-year-old Jesse Gelsinger during his participation in a dose escalation trial administering recombinant adenoviral vector into the hepatic artery for the treatment of ornithine transcarbamylase deficiency dealt a significant setback to the field. It revealed the extent of unethical practices during similar gene therapy clinical trials [11,12,13,14]. Gelsinger’s death was attributed to the vector itself, likely due to induced shock syndrome leading to multiorgan failure [15]. Postmortem analysis showed very little therapeutic effect in terms of altered gene expression [14]. While it is likely that Gelsinger’s response to the viral particles was extremely rare, given the 400 clinical trials with more than 4000 patients involving the same vector [13], his case uncovered a history of ethically questionable practices in gene therapy trials, including lack of informed consent [14] and underreporting of adverse events occurring in trial participants following vector administration [13]. More than two decades after Gelsinger’s untimely death, his memory haunts the field of gene therapy [16]. It has slowed progress toward the treatment of any condition, underscoring the need for safe delivery modalities that are thoroughly vetted in multiple animal models before administration in clinical trials.

In December 2017, nearly 20 years after Gelsinger’s extremely adverse reaction to the viral vector used for his gene therapy, the FDA approved Luxturna™ (voretigene neparvovec-rzyl), an adeno-associated viral vector (AAV) delivery of a normal coding sequence of RPE65 to individuals with biallelic mutations in this gene leading to hereditary retinal dystrophies [17]. At the time of approval, the FDA expressed concern about potential hepatotoxicity, although noting that immune responses in drug-specific trials were mild at all doses [18]. Most of the adverse patient effects in Luxturna™ trials were related to the invasive surgical procedure required to inject the drug at the disease site [19].

The first published proof-of-concept using AAV vectors used a mouse model to replace a mutation that causes progressive retinal degeneration and was published in 1996 [20]. Generally, not thought to cause human disease, the AAV natural genome is a single-stranded DNA molecule containing two inverted terminal repeats (ITRs), which direct genome replication and packaging during the production of the vector [21]. AAVs used for gene therapy lack all viral genes, with the exception of the critical ITRs, and therefore are replication-deficient. These engineered AAVs can cross the cell membrane and deliver DNA to the nucleus, where these transgenes can persist in transformed cells [22]. As the DNA carried in AAV vectors is single-stranded, it must first be converted into a double-stranded transgene in the nucleus before expression can begin [23], a rate-limiting step to the therapeutic benefit. To overcome this obstacle, double-stranded genomes have been packaged in AAV, but that reduces the carrying capacity from about 5 kb to 3.3 kb [23,24]. Potentially the greatest obstacle to widespread AAV usage for gene therapy is the presence of neutralizing antibodies against AAV found in substantial proportions of the human population [25].

Despite the disadvantages of using AAV for gene therapy, the idea of therapeutics that alter gene expression to address diseases is not without merit. The use of plasmid DNA (pDNA) for gene therapy provides an enticing alternative to AAV, given the increased carrying capacity of genetic material and the lower potential for immunogenicity to the vector [26]. Additionally, the same strategy could be accomplished via delivery of RNA; however, compared to RNA, pDNA provides several advantages including easier and less expensive production, longer shelf life [27], less probability of immune system activation [27], and very little risk of genomic integration [28]. pDNA vectors can be engineered to express sequences that are processed in the cell to produce siRNA [29], thus indicating the greater flexibility of DNA as a gene therapeutic.

However, the delivery of plasmids does present its own set of challenges. One of the major obstacles is its large size. pDNA is significantly larger, 5–10 kbp, as compared to siRNA (>30 nucleotides) or even mRNA (up to 1–2 kbp). A second hindrance is the journey of pDNA, which must stay intact from outside of the cell to inside the nucleus. As a result, viable delivery strategies have been limited, and pDNA has been largely abandoned as a therapeutic molecule due to these obstacles. Traditional methods of pDNA delivery are often invasive and/or inefficient, as the size of the nucleic acid prohibits the delivery of therapeutically significant quantities. 

Exosomes, naturally occurring small extracellular vesicles, present an enticing new potential delivery vector for pDNA. Extensive work has been carried out on the exosomal delivery of siRNA and other smaller molecules, but the body of work regarding the delivery of pDNA is limited, likely due to the challenges associated with entrapment, targeted delivery, and therapeutic levels of exogenous gene expression. Here, we review the limited body of work on the delivery of plasmid DNA for gene therapy and potential new technologies using bovine milk and colostrum-derived exosomes that could increase the therapeutic potential of this nucleic acid. 

## 2. Traditional Methods of pDNA Delivery

The potential of pDNA as a vector for gene therapy has been recognized for over thirty years [30]. Currently, more than 60 ongoing clinical trials are registered at clinicaltrials.gov using plasmids therapeutically, with many of those trials focusing on vaccine development (https://clinicaltrials.gov/, accessed on 14 April 2023). With over three decades of research, the progress of pDNA-based therapeutics has not matched that of other forms of gene therapy.

The greatest hurdle for any therapeutic to overcome is targeted delivery to the diseased site without unwanted off-target side effects. Plasmids by themselves do suffer from poor gene transfer efficiency compared to viral vectors, as viruses have evolved complex mechanisms to enter cells and deliver their genetic material to the nucleus [28]. To make its way into the nucleus for therapeutic benefit, pDNA must traverse through the cell membrane, the cytoplasm, and the nuclear membrane, each of which contains mechanisms to exclude free DNA. In certain tissue types, there are other barriers, such as the extracellular matrix of connective tissue [31]. In all cases, the negatively charged plasmid is repelled from the typically negatively charged cell membrane, making crossing difficult and the rate-limiting step for high therapeutic value. These limitations can be overcome by delivery strategies that improve cellular uptake and targeted transfer of DNA.

### 2.1. Naked DNA

In biological systems, the stability of DNA is far superior to RNA [32]. As such, early applications of pDNA for the purpose of gene therapy involved the direct injection of “naked” or unprotected plasmids intraarterially in mice, rats, and nonhuman primates detected high levels of transgene expression in the target muscle tissue [33,34]. Changing the injection site, to the vasculature of the liver, naked pDNA could be delivered to hepatocytes for high levels of exogenous gene expression [35]. Attempts have been made to deliver naked pDNA via more systemic administration routes, such as the tail vein in mice, to achieve expression in the liver; success has been mixed, with some reports of successful hepatic expression [36] and some reports of lack of targeted expression [35]. 

Even after the advent of electroporation (discussed below) to increase cell uptake of pDNA, for some model systems, the use of naked DNA perfused intravascularly resulted in higher levels of exogenous gene expression. Comparing electroporation and hydrodynamic injection of naked DNA for the introduction of cytokine expression, 10,000-fold higher expression in serum was observed with naked pDNA hydrodynamic injection as compared to electroporation, and the level was sustained for 2 weeks after a single injection [37]. 

An inhalable powder has also been produced using naked pDNA and stabilizers such as hyaluronic acid. The addition of hyaluronic acid both prolonged the shelf life of lyophilized formulations of naked pDNA and facilitated inhalation in a murine model, where expression of the transgene could be detected in the lung up to a year after treatment [38]. The administration in this model was intratracheal, and it is not clear if the same efficacy would have been observed if the intranasal route had been used instead. Because intranasal delivery would require the pDNA to traverse the nasal epithelium and likely could alter efficacy, the clinical translatability of such powdered pDNA has not been reported.

Intramuscular injection of naked pDNA has also been successful in human clinical trials, resulting in the approval of plasmid-based gene therapy for chronic lower limb ischemia by the Russian Ministry of Healthcare. In this trial, patients were given two doses of pDNA for expression of VEGF165, and at follow-ups 6 months, 1 year, and 2 years after treatment, therapeutic endpoints were more significantly improved in patients receiving gene therapy than standard treatment [39]. The safety and efficacy of this gene therapy, sold under the name Neovasculgen™, was confirmed in an international post-marketing surveillance survey, prompting the initiation of authorization by the European Medicines Agency (EMA); however, the US FDA has not pursued approval, due to low penetrance of the targeted disease [40].

The use of naked pDNA for either gene therapy or vaccination is not without disadvantages. Only a fraction of the injected pDNA is taken up by the target cells and is subjected to degradation by DNases in interstitial fluid, requiring relatively large doses of 50–200 μg in mouse models [41]. Moreover, the injection techniques used in small animal models, such as hydrodynamic injection utilizing large volumes of liquid or tissue-specific injection requiring invasive surgeries, have not translated well to human studies [42]. To overcome these obstacles, methods have been designed to enhance cellular uptake.

### 2.2. Electroporation

The process of electroporation, also called electropermeabilization, utilizes short, intense electric pulses to transiently introduce instability in the plasma membrane to allow large molecules, such as RNA and pDNA, to enter [43]. In addition to changing membrane permeability, the electric pulse also serves to affect the charge of the DNA molecules, increasing their ability to cross the membrane [44]. The use of electroporation, in comparison to naked DNA injection, may increase cellular uptake by greater than 100-fold [43]. This method has been used to introduce DNA into the skin [45], liver [46], kidney [47], lung [48], and muscle [49]. Because of the presence of keratinocytes in the skin, electroporation can also be used to deliver pDNA for the purpose of vaccination and may itself act as an adjuvant for immunization, evoking the increased presence of immune cells near the site of electric pulse application [50]. During the early days of vaccine development against COVID-19, one of the first publications described a pDNA-based vaccine that elicited T-cell mediated responses in mice and guinea pigs following pDNA delivery of antigen coding sequences and electroporation [51]. The same company responsible for this vaccine, INOVIO, has also developed vaccines against Zika [52], Ebola [53], MERS [54], HPV [55], Lassa fever [56], and tumor-associated antigens [57], although none have made it to market, potentially because of the need for somewhat invasive electroporation for delivery.

Electroporation is not without its limitations and side effects. As with any in vivo method, it is unlikely to reach widespread use, due to the size of the electrodes (about 1 cm) limiting the area of application [43]. Moreover, delivery to some internal organs requires invasive surgical procedures and may result in potential organ damage when exposed to electrical currents [58]. When used to introduce pDNA into muscles, electroporation reduced contraction significantly hours after application and therapeutically sufficient numbers of transformed cells significantly damaged muscle tissue [59]. Additionally, the use of high voltages can damage DNA and may decrease the therapeutic value of whatever is introduced into cells [43]. As with AAV-delivery of genetic sequences, the worse side effects of the drug may not be the pDNA itself or the expression of the transgene, but the damage to access the site of disease. 

### 2.3. Sonoporation

Like electroporation, sonoporation uses ultrasound to increase permeability by creating transient pores in the plasma membranes of target cells allowing for pDNA to enter [60]. Membrane permeability can be increased by using microbubble echo contrast agents [61]. Using a combination of microbubbles and ultrasound, pDNA has been successfully delivered to mice [62], rats [63], and dogs [64]. It does offer an advantage over electroporation, in that pDNA can be delivered to internal organs without the need for surgery [65]. 

First used in animal models in 1991 [66], sonoporation has had limited use due to many disadvantages that also preclude its use in human clinical trials. As with electroporation, the delivery efficiency of pDNA is low in vivo, requiring high doses [67]. Moreover, the application of ultrasound damages cells, altering enzymatic and mitochondrial activity and resulting in apoptosis [68]. In contrast to reports with naked DNA and electroporation, in some cases, expression of transgenes introduced via sonoporation was only detectable for a day following administration before returning to pretreatment levels [69]. When compared directly to naked pDNA delivery, there was no significant difference in the expression of introduced transgenes, although sonoporation resulted in successful gene transfer in more animals per treatment group [70]. In other studies, transgene expression peaks at 4 days post-treatment and can be detected up to 21 days after treatment, with optimized ultrasound exposure, pDNA load, and microbubble concentration [71]. The wide range of success of such experiments indicates the variability of this delivery method and the dependence on optimized conditions for specific tissues. Unfortunately, the degree of ultrasound exposure necessary to achieve more successful gene transfection also caused considerable skin burning at the treatment site [70]. Additionally, injection of some of the microbubble echo contrast reagents has proven to be fatal in murine models [61].

### 2.4. Cell-Penetrating Peptides

Cell-penetrating peptides (CPPs) were first discovered as virally expressed proteins that could penetrate cells and lead to intracellular delivery of cargo, in the absence of receptor interaction [72]. They can be broadly divided into non-cell-specific and cell-specific, with the latter allowing for the targetability of associated cargo. Because some of these CPPs are rich in positively charged amino acid residues, they can be complexed via electrostatic interactions with pDNA [73,74]. However, in vivo efficiency was low due to interaction with serum albumin [73]. CPPs do offer the advantage as they lack cytotoxicity and immunogenicity, but may not demonstrate systemic delivery as detection of the delivered transgenes is only found close to the application site [75]. Modifications of CPPs with poly-histidine or poly-lysine tails have allowed for condensation of plasmids and successful in vivo delivery [76,77]

The use of CPPs permits less invasive modes of administration of pDNA such as topical application. Amphipathic peptide Mgpe9 was used to deliver pDNA following topical application to undamaged skin without toxicity [78]; however, in this study, the transfection efficacy of the peptide did not outperform a lipofection-based competitor, nor was the spread from the application site of transgene expression assessed. 

Another CPP, MPG, a hybrid protein made by fusing gp41 from HIV-1 and the nuclear localization signal from SV40 T antigen, has been used in vaccine development for Hepatitis C. In a murine model, injection of MPG–pDNA complexes subcutaneously in the footpad resulted in the production of serum IgG1 and IgG2a against the structural viral proteins expressed from the pDNA, to a greater extent than naked pDNA [79]; however, the protein-based immunizations for the same viral proteins resulted in higher immune responses than the pDNA. In other vaccine studies, CPP along with cytotoxic T-lymphocyte epitopes have been complexed with pDNA to create a vaccine against human papillomavirus (HPV) and related cervical cancer. In both prophylactic and therapeutic models, the CPP–pDNA complex effectively prevented or inhibited cervical cancer via CD8+ T-cell activation [79,80], but did not achieve clinic translatability. 

The wide variety of CPPs available and the specificity that can be achieved through the selection of cell-type specific targeting moieties holds great promise for other forms of targeted pDNA delivery, but the use of unencapsulated pDNA still requires large doses, >65 μg in small animal models, such as mice, for this method. Moreover, the specificity of CPPs for targeted delivery is still a challenge. For example, while CPPs can be used to enhance the uptake of therapeutic pDNA in tumor cells, gene expression is detected at the same level in non-target tissues, such as the liver and lung [81]. Modifications can be made to improve tumor targeting, but off-target gene introduction could have severe unwanted side effects.

### 2.5. Liposomes

Liposomes are one of the most effective methods for delivering genes because of their remarkable properties in DNA delivery, including improved pDNA stability in vivo, prolonged blood circulation, and low immunogenic reactions. The liposomes fuse with the cell membrane, releasing the plasmids into the cell. The use of encapsulated or entrapped pDNA has overcome the need to use electroporation or sonoporation to induce cell damage to create permeability and has decreased the effective dose of pDNA. Pegylated immunoliposomes, also called Trojan horse liposomes (THLs), can be functionalized for targeted delivery to cells expressing specific receptors without the need for membrane disruption [82]. THLs bearing monoclonal antibodies targeting the transferrin receptor have been used to deliver pDNA to treat Niemann–Pick C1 [83]. The antibody interaction with the transferrin receptor allows the nanoparticles to cross the blood–brain barrier and reach their therapeutic target following intravenous delivery [83]. These antibody-coupled liposomes, also called immunoliposomes, have also been used to deliver therapeutic coding sequences transplacentally, targeting the fetal brain, following intravenous injection to the mother [84]. Furthermore, THLs can be lyophilized and reconstituted, showing no alteration in morphology or efficacy [85], indicating their stability and broadening their pharmaceutical applicability. 

Liposomes themselves do not always have to be modified to ensure target-specific gene expression, as plasmids can be designed to contain tissue-specific promoters [86]. Plasmids, containing prostate-specific promoters, encapsulated by liposomes and administered intravenously, showed tumor-specific expression of the therapeutic gene [87]. The versatility of pDNA to be engineered to express transgenes only in certain cell types can enhance the targetability of the delivery.

Liposomes have demonstrated versatility in loading multiple therapeutic moieties simultaneously for synergistic efficacy. DOTAP (1,2-dioleoyl-3-trimethylammonium-propane) liposomes functionalized with R8-dGR cancer-targeting molecule were loaded with both paclitaxel, a standard-of-care chemotherapeutic, and a CRISPR/Cas9 plasmid targeting hypoxia-inducible factor-1 alpha (HIF-1α) to prevent metastasis of pancreatic cancer [88]. This, and other studies, however, have found that the expression knockdown in vivo with pDNA-delivered CRISPR/Cas9 is significantly less than the efficacy achieved in vitro, an indication that this system has not been fully optimized or cannot achieve similar efficacy inside an organism that is seen in cell culture [89].

In addition to lacking the same level of success in vivo as found in vitro, liposomes have other disadvantages. Liposomes have been found to increase cytokine production by macrophages following systemic delivery, which may have side effects on hematopoiesis, and perturb endoplasmic reticulum homeostasis, also resulting in inflammatory responses [90]. Liposomes have also been found to interact with serum proteins to activate the complement immune system [91]. In addition to these potential immunogenetic effects, the production of liposomes is expensive [92] and modifications such as PEGylation may cause accelerated blood clearance with repeat administration [93]. The specific formulation of liposomes has to be carefully controlled because certain cationic lipids used for the delivery of nucleic acids cause hepatotoxicity and inflammation [94]. Liposome-DNA complexes have also induced the production of interferons, which is also dependent on liposome composition [95].

### 2.6. Other Nanoparticles

Chitosan is a polyaminosaccharide obtained from chitin, which can be found naturally in crustaceans, fungi, and insects [96]. Chitosan can be complexed with pDNA by simple incubation [97]. In cell culture systems, there have been mixed reports as to the efficacy of chitosan-based pDNA delivery. Some cell lines appear to be very sensitive to the associated cytotoxicity, while other cell lines show increased transgene expression compared to other transfection systems [97]. While chitosan has reduced cytotoxicity compared to reagents such as Lipofectamine™, transfection efficiency in vitro is significantly less [98]. Chitosan also has the ability to open up space in tightly packed epithelial layers, such as those found in mucus membranes [99]. Intranasal vaccination using chitosan-complexed pDNA was successfully used in a murine model to vaccinate against RSV [100,101] but did not progress further toward use in humans. 

The transfection efficiency of chitosan-based particles has been improved by coating them with hyaluronic acid (HA), both in vitro and in vivo. Injection of HA-coated chitosan/pDNA complexes directly into tumors resulted in significant suppression of tumor growth [102]. This strategy has also been used to introduce genes into synoviocytes in vitro, a classically hard-to-transfect cell type [103]. 

HA has also been used to increase the transfection efficiency of polyethylene glycol (PEG) and polyethyleneimine (PEI) in vitro. pDNA formulations with HA conjugated to PEI and/or PEG increased transfection efficiency, resulting in high expression of genes delivered via pDNA, and deceased cytotoxicity associated with PEI [104]. However, these nanoparticle approaches have had difficulty making their way into the clinic due to their limited efficiency in vivo compared to viral delivery methods [105], but as discussed previously, viral vectors for introducing exogenous genes present their own set of limitations. 

About two-thirds of clinical trials for pDNA delivery involve these viral vectors because the non-viral methods suffer from a lack of targetability, low efficiency, and rapid clearance before the therapeutic benefit can be achieved [106]. The majority of the remaining one-third involves the delivery of naked pDNA because these nanoparticle-based approaches have not offered real improvement over the viral vectors [106]. The combination of the lack of safety of the most widely used delivery vectors with the inability to successfully create an alternative despite more than 20 years of research, reiterates the need for an alternative pDNA delivery vesicle that offers targetability, biocompatibility, and low toxicity. 

## 3. Exosomes for Plasmid Delivery

Exosomes, a class of small extracellular vesicles, overcome many of the limitations associated with other forms of pDNA delivery. Exosomes are emerging biological nanoparticles whose application in the field of both small molecules and biologics started a dozen years ago [107]. They are generated by the inward folding of the plasma membrane and the formation of multivesicular bodies (MVBs) [108]. Lipid bilayers surround these particles that contain a wide array of proteins, lipids, and nucleic acids, reflecting the biology of their progenitor cells [108]. As natural transport vesicles, exosomes function in intercellular communication by carrying their cargo for various distances within the body [109]. The intrinsic constituents of exosomes increase their time in circulation and minimize immunological responses [110]. Moreover, their enclosure by a lipid bilayer facilitates barrier crossing within the body [111]. Exosomes are produced by many different cell types and are present in most bodily fluids, including blood [112], breast milk [113], saliva [114], and urine [115]. As exosomes hold such promise in the field of pDNA delivery, finding a suitable source seems paramount to expanding the application of these delivery vesicles in the clinic.

### 3.1. Sources of Human Exosomes

Although present in nearly all bodily fluids, in humans, these fluids do not contain enough exosomes to be useful in the pharmaceutical industry. For drug delivery, exosomes of human origin are typically purified from conditioned cell culture media produced in bioreactors. From this media, the maximum yield approaches 10^13^ particles per liter [116]. Conditioned cell culture media containing exosomes can be produced from a variety of types of cell lines, including adipose stem cells [117], bone-marrow-derived mesenchymal stem cells [116], Wharton’s jelly-derived stem cells [118], and numerous human cell culture lines, both cancerous and normal [119,120]. While there are several commercially available exosome isolation reagents, such as ExoQuick™ (System Biosciences, Mountain View, CA, USA) and Total Exosome Isolation Reagent (Thermofisher, Vilnius, Lithuania); however, for large-scale isolation, the most often used procedure is ultracentrifugation [116,117]. Each research group uses a slightly different protocol, likely based on their own optimization but the general procedure is the same: low-speed centrifugation to clear the media of cellular debris followed by sequential ultracentrifugation to pellet exosomes [121]. Further purification steps may include ultrafiltration and size-exclusion chromatography [122].

Following isolation, exosomes are typically validated to exclude the presence of another subpopulation of extracellular vesicles, such as microvesicles or apoptotic bodies. These characterization steps include the determination of size by dynamic light scattering or electron microscopy, detection of exosomal surface protein markers, and surface charge, or zeta potential [123]. The precise range of sizes that encompasses exosomes is not completely agreed upon, but 30–150 nm diameter is generally accepted [124]. Classic exosomal protein markers include tetraspanins such as CD9, CD63, and CD81 and MVB formation proteins such as Alix and TSG101 [124]. Exosomes carry the integrin-associated transmembrane protein CD47 [125], which is considered anti-phagocytic [126] and may increase blood circulation time. While isolation of highly purified exosomes is possible from conditioned cell culture media, for clinical trials or large-scale production, the volume of media that would need to be generated is astounding, substantially increasing the cost and underscoring the need for another more viable source.

### 3.2. Bovine Milk as an Exosome Source

Bovine milk and colostrum are the most abundant known sources of exosomes, yielding a minimum of 3 × 10^14^ and 4 × 10^15^ particles per liter, respectively [127]. Isolation of exosomes from these sources is highly cost-effective and easily scalable to drug development production levels. According to the US Department of Agriculture, as of 1 January 2023, there are nearly 10 million dairy cows in the US alone (https://www.nass.usda.gov/Newsroom/2023/01-31-2023.php, accessed on 28 April 2023), making bovine milk products readily available for large-scale isolation of exosomes. Bovine colostrum is the milk produced in the first few days after calving, which provides support for newborn calves both nutritionally and immunologically [128]. Moreover, mother cows produce an excess of colostrum compared to their calves’ dietary needs [129], and the use of bovine colostrum as both a dietary and health supplement has been practiced for centuries [130].

Bovine colostrum powder is standardized and commercially available. Dried colostrum powder can be rehydrated and used to isolate exosomes in a manner similar to that described for conditioned cell culture media with sequential steps of ultracentrifugation followed by additional purification [131]. Bovine milk and colostrum-derived exosomes are of a similar size as cell-culture-derived exosomes, with similar zeta potential [132]. Multiple batches of colostrum produce consistent quantities of exosomes with similar particle size distributions and an abundance of exosomal markers [131]. Like exosomes derived from other cell sources, bovine milk, and colostrum exosomes have surface-associated CD47, an antiphagocytic signal, increasing circulation time [126]. They also have other characteristic exosomal markers, including tumor susceptibility gene 101 (TSG-101), tetraspanins CD81 and CD63, and endosomal sorting complexes required for transport (ESCRT)-associated proteins such as Alix [131].

## 4. Advantages of Bovine Milk Exosomes for Plasmid DNA Delivery

### 4.1. Lack of Systemic Toxicity, Immunotoxicity, and Immunogenicity

The use of bovine-derived exosomes in either rodents or humans represents the administration of protein-derived therapeutics across species, which raises the question of safety and potential for immune response. Oral treatment of female Sprague-Dawley (SD) rats with bovine milk exosomes showed no alteration in kidney or liver biochemical profiles following single or 15-day daily dosing regimens [133], indicating a lack of systemic toxicity. In a longer study, wild-type (WT) mice given bovine colostrum exosomes orally three times per week for 4 weeks generally showed no differences in kidney and liver function tests as compared to untreated mice; a nominal modulation of parameters such as total bilirubin, calcium, and sodium concentrations was observed, but the levels were within physiological normal ranges for mice [132].

Immunotoxicity refers to the aberration of the function of both local and systemic immune systems as a result of exposure to toxic substances and may result in a state of immune suppression or autoimmunity [134]. In female SD rats, 6 h of exposure or 15 days of daily dosing with bovine milk exosomes did not result in neutropenia or lymphopenia [133]. A similar result was observed following dosing at a frequency of 3 times per week for 4 weeks in WT mice [132,135].

The immunogenicity of a therapeutic is its potential to trigger an unwanted immune response against the drug itself [136]. There are many ways to assess the extent and specificity of such immune responses to potential therapeutics. Ascertaining serum cytokine levels can determine if an immune response is being mounted against a particular therapeutic agent. Individual cytokines often have multiple and even paradoxical roles, both pro- and anti-inflammatory, depending upon the receptor they bind to and the type of cell they activate. Therefore, it is not possible to tell exactly what type of immune response is elicited as a result of exposure. Still, any induction of cytokine expression as a result of treatment would indicate a potential immunogenic response.

To determine the potential immunogenicity of bovine exosomes in other species, a panel of species-specific cytokines was analyzed following treatment. Interleukin (IL)-1α and IL-1β are both pro-inflammatory cytokines whose presence is upregulated by a variety of stimuli, including tissue damage and the presence of pathogens [137,138]. IL-6 participates in a diverse array of immune responses, including promoting acute-phase reactions [139]. Interferon-g (IFN-γ) is also upregulated during inflammatory responses, such as the detection of foreign peptides by antigen-presenting cells [140]. IL-2 and IL-12 both activate T-cell subpopulations. IL-2 is produced upon recognition of foreign peptides by activated B-cells leading to T-cell activation [141], while IL-12 is produced by dendritic cells in response to antigen recognition [142]. IL-4 and IL-13 regulate allergic inflammation [143], while IL-5 functions in similar inflammatory reactions via activation of eosinophil maturation and release [144]. SD rats treated with either a single dose for 6 h or daily doses for 15 days of bovine milk exosomes showed no significant elevation of serum concentration of classically pro-inflammatory cytokines (Figure 1A), T-cell activating cytokines (Figure 1B), or cytokines increased during allergic responses (Figure 1C) [133]. This finding has been confirmed by independent researchers, who have also demonstrated that exposure to bovine milk exosomes does not illicit anaphylaxis or increased serum histamine levels following multiple doses or serial injections [145].

Determination of the presence of bovine exosome-specific antibodies would be the ultimate test of immunogenicity; however, these tests are costly and difficult due to a high degree of homology between bovine and other mammalian exosomal proteins. The lack of cytokine elevation after both short- and longer-term exposure, coupled with the lack of hypersensitivity responses, provides strong evidence for the immunological safety of bovine exosomes for use in cross-species pharmaceutical applications.

### 4.2. Administration by Multiple Delivery Routes

Unlike some of the other methods mentioned previously, such as electroporation or sonoporation, delivery of pDNA by bovine exosomes does not require invasive procedures such as surgery to target a specific organ or region of the body. Simply changing the administration route, all of which are minimally invasive, can result in the delivery of exosomal formulations to specific organs. The biodistribution of bovine milk exosomes delivered orally versus intravenously results in different patterns of accumulation, with i.v. delivery predominating to the liver and oral delivery resulting in more evenly distributed exosomes in the liver, lung, kidney, pancreas, spleen, ovaries, colon, and brain [133]. Bovine milk exosomes delivered orally have a high bioavailability, which makes them ideal for the development of oral therapeutics [146].

### 4.3. Functionalization for Targeted Delivery

In addition to using different routes of administration to achieve organ-specific targeting, bovine exosomal formulations can be modified to enhance organ-specific delivery of payload and direct cell-specific targeting. For example, folate receptor alpha (FRα) is highly expressed on many tumor cells as compared to healthy tissue (Figure 2A) and can be used as a targeting moiety to deliver therapeutics preferentially to the tumor [147,148,149]. Bovine milk and colostrum exosomal formulations can be functionalized with folic acid (FA) to target highly expressed FRα. Briefly, following bovine milk or colostrum exosome isolation via ultracentrifugation, purified exosomes can be incubated with specified concentrations of activated FA followed by brief ultrafiltration [131,132,133]. The result is the attachment of FA via standard EDC [1-ethyl-3-(-3-dimethyl aminopropyl) carbodiimide hydrochloride) and NHS (*N*-hydroxysuccinimide esters) [132]. The addition of FA does not alter the physical characteristics of exosomes, such as particle size or surface charge [131,132]. Hyaluronic acid has also been used to modify bovine milk exosomes for the purpose of tumor targeting, although it has only been explored in vitro [150].

## 5. Mechanisms of Loading Exosomes with Plasmid DNA

The use of exosomes to deliver large biological molecules such as plasmids has not been as extensively studied as using these nanoparticles to deliver small molecules or smaller nucleic acids such as siRNA and mRNA. The strategies for loading pDNA onto/into exosomes are similar to strategies used for other cargo, including: electroporation, sonication, exosome–liposome hybrids, and transfecting exosome-producing cells.

### 5.1. Electroporation

Electroporation involves the application of an electrical pulse to exosomes, producing transient pores that allow for molecules like DNA to enter the lumen of the vesicle [151]. Using this method, the CRISPR/Cas9 genome editing platform encoded on a single plasmid has been loaded into human mesenchymal stem-cell-derived exosomes and delivered both in vitro and in vivo for the treatment of pancreatic cancer [152]. Additionally with exosomes derived from ovarian cancer cells, targeted delivery of CRISPR/Cas9 construct containing plasmids was administered both intratumorally and intravenously and delivered cargo specifically to tumors derived from the same cells [153], demonstrating the trafficking ability of these particles.

Using electroporation, exosomal entrapment of pDNA is low and in some cases has been found to distort the nucleic acid to the point that gene expression from the plasmid was not detectable in vitro [151]. Multiple investigators have reported the potential of electroporation to change the morphological characteristics of exosomes [154,155]. Linear double-stranded DNA has been found to have a higher loading efficiency than circular plasmids, and using this technique, linear DNA longer than 750 base pairs was not efficiently encapsulated [151]. In recent reports, the efficacy of electroporation in nucleic acid loading of exosomes has drawn scrutiny, with some researchers reporting the formation of insoluble precipitates that may be inflating the entrapment efficiency of exosomes following the application of charge [156].

### 5.2. Sonication

Sonication has also been reported as a means of loading exogenous cargo onto exosomes. This process also involves the disturbance of the lipid bilayer to allow for the entrance of drugs like DNA into the interior of the particle. Sonication has been used to load DNA into exosomes derived from human cell lines, but the DNA loaded was single-stranded and significantly smaller than a plasmid, only 25 nucleotides in length [157]. Sonication does seem to produce less aggregation of nucleic acids as compared to electroporation [157]; however, the sonication itself increases the size of exosomes even in the absence of drug cargo [158,159], which could potentially alter the trafficking pattern of the exosomes. Most importantly, sonication is a known method for creating DNA fragments [160] and therefore may not be suitable for loading large DNA molecules like plasmids onto exosomes.

### 5.3. Hybrid Exosomes

To improve the encapsulation and integrity of DNA, exosome–liposome hybrids have also been used to deliver pDNA, which do not require electroporation for loading. Applications include delivery of the CRISPR/Cas9 system to many cell types, such as stem cells, which cannot be transfected by liposomes alone [161], and chondrocytes [162]. All published applications of these hybrids have been in vitro, and the in vivo efficacy remains to be assessed.

### 5.4. Transfection of Exosome-Producing Cells

A novel strategy for the loading of gene-encoding DNA into exosomes has been to transfect the cells generating exosomes with plasmids containing sequences of interest and isolate exosomes released from these cells, thereby encapsulating DNA at the point of biogenesis. This strategy has been used to deliver coding sequences for catalase. Macrophages were transfected ex vivo with pDNA encoding for catalase and then injected into mice for efficacy studies. Exosomes released from these macrophages contained detectable amounts of DNA for the transgene, indicating that the DNA has been packaged as part of the production of the exosome itself [163]. In this study, the transfected macrophages contained significantly more coding sequences for the introduced transgene than non-transfected cells, but no comparison was performed with other loading methods, such as electroporation. This type of cargo loading also requires an additional step, i.e., transfecting the parental cells, which may limit the scalability and feasibility of large-scale production.

### 5.5. Exosome-Polyethyleneimine Matrix (EPM)

Despite the promise of exosomes for drug delivery, due to their natural trafficking ability and targetability to specific cell populations or organs in the body, the work in this arena has not kept up with the use of exosomes for the delivery of smaller nucleic acids, such as siRNA or mRNA. Part of the difference lies in the misperception that exosomes are not well suited for the conveyance of large biomolecules such as plasmids. The loading efficiency of plasmids into exosomes has been described as low due to the small size of exosomes and the application of exosome-loaded plasmids as a viable therapeutic questioned [164,165].

#### 5.5.1. Entrapment of pDNA within the EPM

Using bovine milk and colostrum exosomes and classic transfection reagents, our research group has pioneered the development of a nanoplatform for the delivery of nucleic acids, namely the exosome polyethylenimine (PEI) matrix, or EPM, which overcomes the limited loading capacity of other methodologies. PEI, a polycation, has classically been used as a transfection reagent for pDNA and siRNA, allowing for the packaging of nucleic acid, delivery across the plasma membrane, and the breaking of endosomal particles via the proton sponge effect [166]. The same properties of PEI can be extorted to mediate the interaction between negatively charged pDNA and negatively charged bovine exosomes via electrostatic interactions.

This nanomatrix demonstrated a high entrapment efficiency of pDNA. While exosomes alone are unable to entrap pDNA, likely due to the repulsion of similarly charged molecules, the EPM is able to entrap more than 95% of pDNA at loads of both 2 μg and 10 μg (Figure 3), compared to just over 37% entrapment by PEI alone. Higher entrapment has correlated with high transfection both in vitro and in vivo [131]. Nucleic acid entrapment in the EPM was also compared to electroporation and commercial reagents, with the EPM demonstrating >90% entrapment, compared to <5% entrapment with electroporation and approximately 35% entrapment with ExoFect™ (System Biosciences) [131].

#### 5.5.2. Functionalization of the EPM 

Bovine exosomes as part of this delivery system can also be functionalized with FA, creating the FA-EPM. Compared to the unfunctionalized formulation, FA-EPM accumulates in tumors significantly more, while localization to the normal lung (with low levels of folate receptors) is approximately the same with both formulations (Figure 2B). Additional targeting ligands are currently under investigation, including lactoferrin [167] for targeting inflamed cells in chronic lung inflammatory diseases such as COPD.

## 6. Therapeutic Application of pDNA via Bovine Milk/Colostrum Exosomes

Despite the strong evidence in favor of using bovine milk and colostrum exosomes as part of a platform for pDNA delivery, our group may in fact be the only one that has published in this area. Table 1 summarizes the different genes we have introduced both in vitro and in vivo, the size of the plasmid carrying the coding sequence, and the therapeutic potential of exogenously expressing each gene. For proof-of-concept, pDNA encoding green fluorescent protein (GFP) has been delivered in vitro to lung cancer cells, demonstrating the transfection capacity of the EPM and its superior performance to PEI alone. When used to transfect A549 lung cancer cells, EPM carrying pDNA bearing the GFP coding sequence produced nearly 4 times more GFP-positive cells than PEI alone [131]. In p53-null lung cancer cell line H1299, the coding sequence for p53 was introduced via EPM and PEI alone. EPM-delivered pDNA resulted in nearly 3 times more p53 expression compared to PEI alone [131]. Additionally, expression of p53 via EPM also increased the sensitivity of both lung cancer cell lines to paclitaxel, a standard-of-care chemotherapeutic, as assessed by both cell survival and colony-forming assays [131]. To demonstrate the efficacy of this delivery system in vivo, EPM-delivered pDNA encoding p53 was administered to p53-null mice intravenously for 6 days, with tissues harvested 24 h after the last dose. Expression of p53 was detected in all tissues, with the highest levels found in the lung and spleen (Figure 4) as assessed by both qPCR and Western blot. FA-functionalized EPM was used to deliver the same plasmid to tumor-bearing nude mice, also resulting in detectable p53 expression in the tumor as assessed by western blot [131].

The EPM has also been used to deliver pDNA for additional purposes besides cancer therapeutics. The COVID-19 pandemic beginning in early 2020 underscored the urgency for the development of new systems with which to study viral therapeutics. Very few labs in the US have access to BSL-3 facilities necessary to handle live viruses, such as SARS-CoV-2, the etiological agent of COVID-19. Using the EPM nanoplatform, multiple pDNA carrying coding sequences for three key SARS-CoV-2 antigens were delivered in vitro and in vivo for the purpose of developing a lower containment level model to screen gene-based antiviral therapeutics, such as siRNA, and potential vaccine development [167]. Following intravenous administration, expression of antigens could be detected in the lung and spleen, even 15 days after administration [167].

## 7. Conclusions and Future Perspectives

The identification of genetic anomalies leading to disease has prompted an entirely new approach to the development of therapeutics that directly address aberrant patterns of disease-related gene expression. Although DNA was first identified as the nucleic acid responsible for harboring inheritable genetic material, the use of DNA, specifically in the form of non-viral associated vectors, has lagged behind research with other, smaller molecules such as siRNA and mRNA. The size of pDNA, which contributes to poor loading efficiency on vectors for delivery, has likely contributed to the lack of successful drug development. Many attempts have been made to deliver pDNA to the disease site for therapeutic benefit, including electroporation of the target tissue, sonoporation applied in a similar manner, and delivery via liposomes and other nanoparticles, but no marketable therapeutic has emerged.

The use of bovine milk and colostrum exosomes holds great promise for the future of drug delivery due to their natural trafficking ability, safety, and ability to be modified to target specific organs or cell types. We have demonstrated that these nanoparticles can be loaded with plasmids carrying various coding sequences as mediated by PEI and deliver those coding sequences for exogenous gene expression to a variety of tissue types, with “zip-code-like” specificity when exosomes are functionalized. The EPM platform has overcome many of the shortcomings associated with other pDNA delivery systems, such as invasive administration and low pDNA entrapment. The possibility of plasmids that this system can entrap is profound and the number of diseases that are potential therapeutic targets can be expanded as our understanding of receptors or surface-associated proteins specific to these diseases expands. As a result of the wide availability of bovine milk and colostrum and the abundance of exosomes within these sources combined with cost-effectiveness, they seem practical sources for harvesting sufficient exosomes for clinical trials and pharmaceutical production. Although initial evaluations of immunotoxicity and immunogenicity have been completed, a more thorough examination of any potential alteration or elicitation of immune response and subchronic/chronic toxicity would be needed before declaring bovine exosomes as a safe platform for drug delivery.

## 8. Patents

The authors have filed an international patent application (PCT) based on part of the results reported in this paper.

## Figures and Tables

**Figure 1 pharmaceutics-15-01832-f001:**
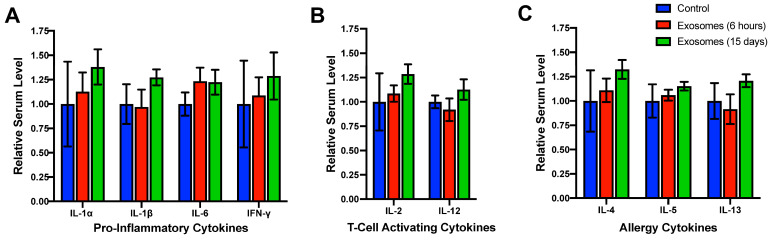
Lack of modulation of species-specific cytokines following exposure to bovine milk exosomes. Female Sprague–Dawley rats (5–6 weeks old) were treated by oral gavage with a single dose or daily doses for 15 days of bovine milk exosomes (25 mg/kg b. wt.). Six hours after the final dose, animals were euthanized and serum was collected for assessment of cytokine concentration using Bio-Plex cytokine T_H_1/T_H_2 assay. Animals exposed to bovine exosomes were compared to vehicle-treated controls. Cytokines assayed included those related to (**A**) classic inflammation pathways; (**B**) T-cell activation; and (**C**) allergic responses. Adapted with permission from Munagala et al., 2016 [133].

**Figure 2 pharmaceutics-15-01832-f002:**
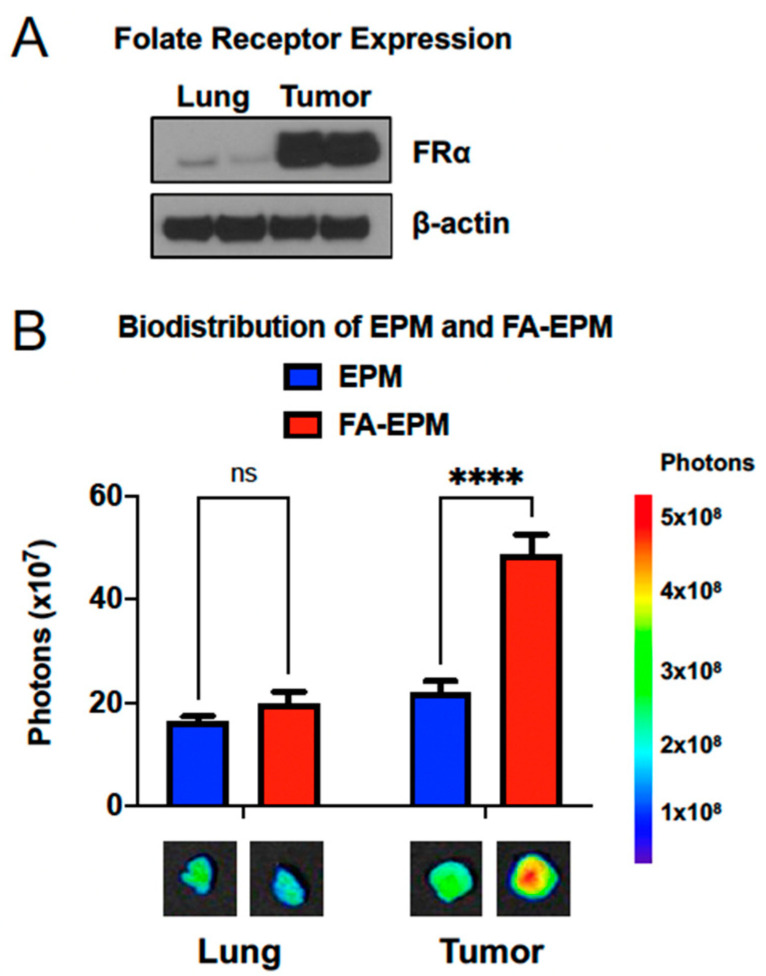
Abundance of folate receptors and biodistribution of exosomal formulations with and without folic acid functionalization. (**A**) Detection of folate receptors in normal mouse lung and in A549 subcutaneous lung tumor xenografted in nude mice. Cells and tissue protein lysates were analyzed by western blot. FR-α, folate receptor α. (**B**) Biodistribution of bovine colostrum exosomes and EPM, with and without FA-functionalization of exosomes. Athymic nude mice were inoculated with A549 human lung cancer cells subcutaneously. When tumors reached 300–400 cm^3^, AF750-labeled bovine colostrum exosomes complexed with polyethylenimine (EPM), with and without folic acid functionalization, were administered intravenously. Animals were euthanized after 4 h and select tissues were imaged ex vivo using an advanced molecular imager (AMI1000). While FA-functionalization of EPM only marginally altered accumulation in the lung, folic acid targeting of folate receptors expressed on the tumor significantly increased accumulation (ns = not significantly different, **** *p* < 0.0001). Warmer colors in ex vivo organ images indicate higher fluorescent signal intensity. Data were analyzed via 2-way ANOVA and post hoc comparison of column means. Adapted with permission from Munagala et al., 2021 [131].

**Figure 3 pharmaceutics-15-01832-f003:**
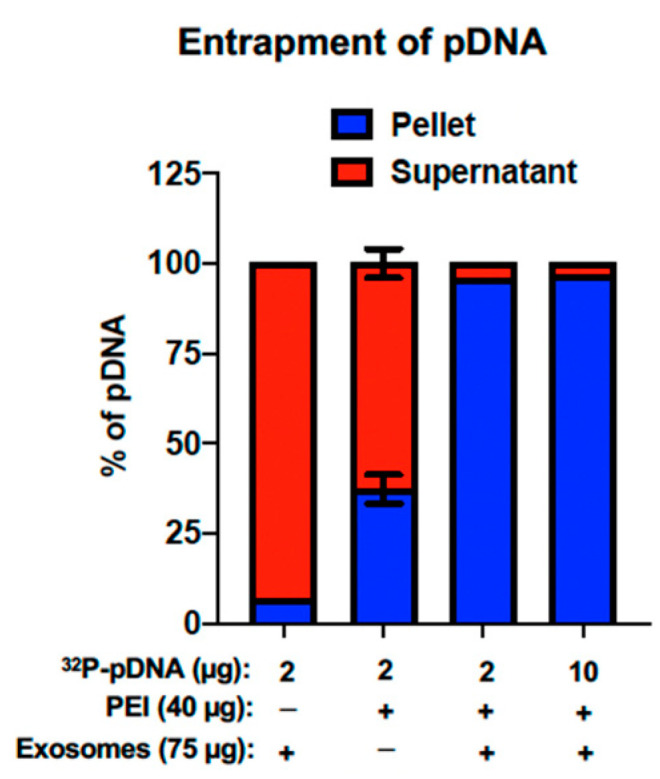
Entrapment of plasmid DNA by the EPM. The entrapment efficiency of the EPM was assessed using plasmid DNA at concentrations of 2 μg and 10 μg per reaction. Briefly, bovine colostrum exosomes (Exo) alone, polyethylenimine (PEI) alone, or Exo + PEI (EPM) were incubated with 2 or 10 μg of pDNA (including ^32^P-labeled pDNA as tracer) and the resultant complex harvested by PEG 2K precipitation. After harvesting, both the pellet and the supernatant were spotted, and ^32^P-signal was measured by Packard Imager. Adapted by permission from Wallen et al. 2022 [167].

**Figure 4 pharmaceutics-15-01832-f004:**
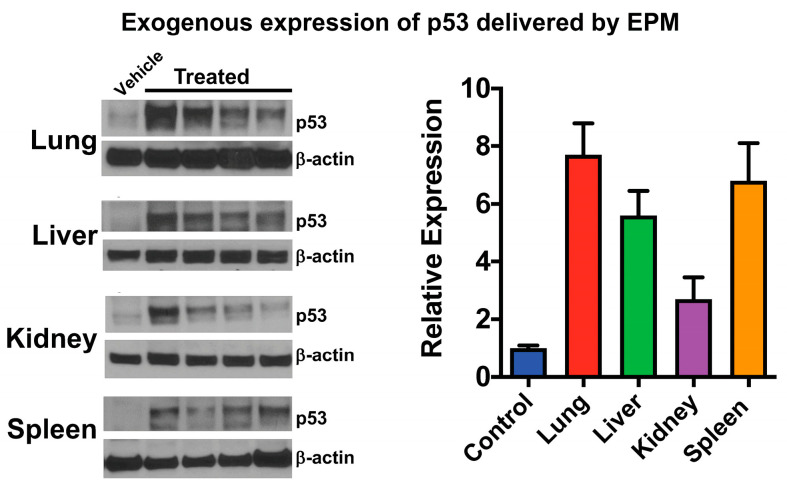
Delivery of plasmid DNA by the EPM. EPM loaded with pDNA deliver the coding sequence for p53 in vivo to p53-null mice following intravenous delivery. Mice were treated for 6 days (0.6 mg/kg pDNA). Tissues were harvested 24 h after the last dose and p53 levels were assessed via western blot. All tissues analyzed contained detectable p53 expression, with the highest levels being in the lung and spleen. Adapted with permission from Munagala et al., 2021 [131].

**Table 1 pharmaceutics-15-01832-t001:** Genes delivered by EPM technology.

Gene Delivered	Size of Plasmid (kb)	Therapeutic Application	In Vitro Application	In Vivo Application	Reference
emGFP	6.2	Proof-of-concept	✓	✓	[131]
p53	6.4	Cancer treatment	✓	✓	[131]
SARS-CoV-2 Spike protein	9	Development of antiviral therapeutics including vaccines	✓		[167]
SARS-CoV-2 Spike protein subunit 1	7.2		✓	✓	[167]
SARS-CoV-2 nucleocapsid	6.5		✓	✓	[167]
SARS-CoV-2 replicase	8		✓	✓	[167]
CRISPR/Cas9 and sgRNA ^ab^	9.3	Genome editing	✓		

^a^ Unpublished data. ^b^ sgRNA = single guide RNA.

## Data Availability

No new data were created for this manuscript.

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
