# Peer review of "Exosomes as an Emerging Plasmid Delivery Vehicle for Gene Therapy"

_pharmaceutics, 2023, doi:10.3390/pharmaceutics15071832_

Round 1

Reviewer 1 Report

Regarding the manuscript (pharmaceutics-2432211) entitled:

Exosomes as an Emerging Plasmid Delivery Vehicle for Gene Therapy

Comments to the Author

General comment

The manuscript describes the various methods for fabricating bioactive molecule-loaded fibrous scaffolds. I have some few comments to be considered before publication:

1.      Previously published reviews like

https://pubs.rsc.org/en/content/articlelanding/2021/NR/D0NR07622H

https://www.mdpi.com/1999-4923/15/2/598

2.      authors should compare this review with previously available literature, what is lacking in the previous reviews and what information you will be exploring this review.

3.      Isolation methods and physicochemical characterization of exosomes. For instance, the papers  https://doi.org/10.3390/nano11061481 10.1016/j.addr.2012.06.014 provided information on these with detailed tables.

4.      drug loading of exosomes should be covered.

5.      Fabrication Methods: Diagram for each technique and parameters affecting the process should be added.

6.      Limitations of marketed products and future aspects should be presented as the final section.

Reviewer 2 Report

The title of the manuscript is good. English language has good quality. Figures and Tables are acceptable.

1. In page 1, line 26

Why the authors have inserted "cancer" in keywords?

2. Page 1, line 29-37

Why the sentences of this part have no proper reference?

3. Page 1, line 38-39

Please add suitable reference at the end of this sentence

4. Line 55-59 in page 2 need proper reference

5. Line 60 in page 2 has multiple references, please reform it

6. Line 67-70 in page 2 need suitable reference

7. Page 2 and 3, line 94-103

Some sentences doesn't have proper references.

Pease consider proper references.

On the other side, some sentences have multiple references.

Please reform all of them.

8. In the part "Introduction" the authors have not explain enough information about exosome and its application in plasmid delivery vehicle.

Please explain why?

9. Line 106-108 in page 3 needs proper reference. Please consider it.

10. Line 108-110 is better to be reformed and turn into a sentence without question. Please

turn it into a declarative sentence.

11. Page 4, line 123 needs proper reference

12. Page 4, line 195-196 needs proper reference

13. Line 334-341

This part contains no reference. Why?

14. The title of Figure 1 contains some sentences which belong to other part of manuscript. For instanse, line 435-439 does not belong to the title of this figure. Please reform the title of mentioned figure.

15. All of the manuscript, there are some sentences without proper reference. Please add suitable reference at the end of all of the sentences of manuscript (except well-established sentences and results of present study)

16. The title of each figure should be reconsidered. Title of each figure should represent details of figure clearly. Other extra explaination (including information abour material and methods or data about other studies) should not be mentioned in the title of figure

Please reform the title of each figure based on remarked note

17. Why the mamuscript does not have the part "Conclusion"?

18. Why the authors have not explained about the comparison between bovine milk and colostrum exosomes and other types of exosomes, e.g. bone marrow derived exosomes?

19. Please check and adjust the "Reference list" based on the regulations of reference list of journal. (Titles, doi, the name of journal and ... )

Round 2

Reviewer 1 Report

no comments